# Depression Levels Influence the Rate of Asthma Exacerbations in Females

**DOI:** 10.3390/jpm11060586

**Published:** 2021-06-21

**Authors:** Papaporfyriou Anastasia, Tseliou Eleni, Mizi Eleftheria, Ntontsi Xenia, Papathanasiou Eygenia, Souliotis Kyriakos, Dimakou Katerina, Bakakos Petros, Loukides Stelios, Hillas Georgios

**Affiliations:** 1Department of Respiratory Medicine, University of Athens Medical School, “Sotiria” Chest Diseases Hospital, 11527 Athens, Greece; dranastp@gmail.com (P.A.); petros44@hotmail.com (B.P.); 2Department of Respiratory Medicine, University of Athens Medical School, ‘Attikon’ Hospital, 11527 Athens, Greece; elentsel@gmail.com (T.E.); xenia-1990@hotmail.com (N.X.); evgjenaki11@yahoo.com (P.E.); loukstel@med.uoa.gr (L.S.); 3Department of Critical Care and Pulmonary Services, Medical School, National and Kapodistrian University of Athens, Evangelismos Hospital, 11527 Athens, Greece; eleftheria.mizi@yahoo.com; 4Department of Social and Educational Policy, University of Peloponnese, 20100 Corinth, Greece; info@ksouliotis.gr; 5Department of Pulmonary, “Sotiria” Chest Diseases Hospital, 11527 Athens, Greece; kdimakou@yahoo.com

**Keywords:** asthma, control, exacerbations, depression, anxiety, gender

## Abstract

**Background****:** Anxiety and depression are common psychological disturbances among asthmatic patients. The aim of the present study is the assessment of anxiety and depression in asthmatic patients and their correlation with symptoms control level and number of exacerbations per year. **Methods****:** One hundred patients with asthma diagnosis, according to the Global Initiative for Asthma (GINA), aged > 18 years old, having a stable disease, were included. Emotional status was evaluated using the Hospital Anxiety Depression Scale (HADS). Patients were followed up for a year to assess the number and severity of exacerbations. **Results****:** Most of our patients were female (58%), middle-aged (mean = 54 ± 13), and married (81%), with low frequency of smoking habits (smokers, ex-smokers and non-smokers were 26%, 30% and 37%, respectively) and low levels of both anxiety and depression [median (interquartile range (IQR)) = 4(2) and median (IQR) = 4(2), respectively]. At the low and moderate level of the depression subscale, female patients experienced asthma exacerbations more frequently compared to male patients (adjusted Incidence Rate Ratio (aIRR) = 4.30; 95% Confidence Interval (CI): 1.94–9.53 and aIRR = 1.82; 95% CI: 1.07–3.13, respectively). **Conclusions.** Clinicians should evaluate asthma patients for depression, as gender differentially influences outcomes among those with low and moderate levels of depression, with female asthmatics presenting more frequent exacerbations.

## 1. Introduction

Asthma is a chronic airway disease that encompasses a patient’s lifetime, provoking physical symptoms, as well as significant mental and social problems. The aim of asthma therapy is to reach and maintain disease control, and minimize symptoms, daily activity limitation, and the risks for life-threatening exacerbations and long-term morbidity [1]. Asthma control is sometimes difficult to achieve because of the interaction of different causes related to the disease itself, treatment (i.e., inadequate treatment, inhaler device) and patient characteristics (i.e., socio-demographic factors, adherence, co-morbidities) [2].

Among co-morbidities, psychopathological ones are frequently observed in asthma patients, resulting in a significant negative impact on the quality of their life [3]. Emotional disorders, including anxiety and depression, are more prevalent in asthma compared to the general population [4,5]. Particularly, the prevalence of depression or depressive symptoms in asthma patients ranges between 11% and 13% [5,6]. Moreover, clinical data has shown that the presence of psychiatric and psychological symptoms is associated with increased asthma symptoms, health service use and costs [3,7,8]. More specifically, in the Severe Asthma Research Population (SARP), patients with insomnia, anxiety and depression had a 2.4-fold increased risk for poor asthma control and a 1.5 higher risk for healthcare utilization, suggesting a significant impact of these conditions on asthma-related outcomes [9]. Available evidence from a meta-analysis of prospective studies showed that the presence of depression may make someone susceptible to asthma, rather than that asthma precedes and predisposes to the development of depression [10].

The aim of the present study is to investigate the impact of anxiety and depression on the control level and exacerbations rate during a one-year-follow-up, in a cohort of Greek patients with asthma.

## 2. Materials and Methods

### 2.1. Study Patients

Participants were recruited from pulmonology clinics of Attikon Hospital, Evaggelismos Hospital and Sotiria Chest Hospital, all of them sited in Athens, Greece. The study protocol was approved by Local Scientific Committees of the three hospitals and all patients provided written informed consent. One hundred patients with asthma diagnosis according to GINA, aged > 18 years old, having a stable disease for at least a month were included. All patients were optimally treated according to GINA guidelines [1]. Two experienced clinicians obtained the personal details and family history of all participants. Patients with a diagnosis of other respiratory disease, concomitant malignancy or severe heart, liver, renal or collagen disease were not included in the study. Emotional status was evaluated using the Hospital Anxiety Depression Scale (HADS). Patients were followed up for a year by telephone to assess the number of exacerbations. During follow-up, asthmatic exacerbations were defined according to American Thoracic Society (ATS) and European Respiratory Society (ERS) Task Force [11]. According to this Task Force, exacerbations of asthma were events that were characterized by any of the following: (a) deterioration of symptoms outside the patient’s usual range of day-to-day variability, (b) deterioration of lung function, (c) increased use of rescue medication for at least 2 days, (d) use of systemic corticosteroids or an increase from a stable maintenance dose for at least 3 days, and (e) hospitalization or emergency department visit because of asthma symptoms, requiring systemic corticosteroids.

### 2.2. HADS Questionnaire

Participants were asked to fill out the self-reported HADS questionnaire [12]. The HADS is a 14-item scale that generates ordinal data. Seven of the items relate to anxiety and seven relate to depression. Each item is rated on a 4-point scale: 0 indicating not at all; 1, sometimes; 2, often; and 3, all the time. This gives a maximum subscale score of 21 for anxiety and depression, respectively (HADS-A and HADS-D). A total subscale score of >8 points out of possible 21 points, indicates significant symptoms of anxiety or depression. HADS gives clinically meaningful results as a psychological screening tool in clinical group comparisons and correlation studies with several aspects of the disease and the quality of life [13].

### 2.3. Statistical Analysis

Relative and absolute frequencies were used to describe categorical variables, whereas means with standard deviations and medians with interquartile ranges were used to describe normally and non-normally distributed continuous variables, respectively. The normality of the distributions was assessed using Kolmogorov–Smirnov tests.

Negative binomial regression analysis was used to examine the relationship among anxiety and depression levels and the number of asthma exacerbations experienced in a year. Interaction effects between the gender and depression subscale was entered into the same model to examine whether the effect of depression on asthma exacerbations differentiates by gender. All continuous predictors were centered at their means. All analyses were conducted using Stata v13.1.

## 3. Results

Of the sample of 100 patients who participated in the study, 58% were female and middle-aged (mean = 54 ± 13). The majority had fulfilled mandatory education (47.0%) and were married (81%). Half of the sample had been diagnosed with asthma at least 13 years ago (median = 13, IQR = 13). Regarding their smoking habits, 26% of the patients were smokers, whereas the relative percentages for ex-smokers and non-smokers were 30% and 37%, respectively. Participants displayed low levels of both anxiety and depression (HADS-A: median (IQR) = 4(4) and HADS-D: median (IQR) = 4(2), respectively). The socio-demographic and clinical characteristics of the sample are presented in Table 1.

The results of the negative binomial regression indicated that there was a significant depression–gender interaction after controlling for anxiety levels [IRR (Incident Rate Ratio) = 0.72; 95% CI: 0.58–0.90]. To examine the nature of the interaction, the depression x gender interaction effect was probed at 1 SD below the mean of the depression subscale (defined as “low”), at the mean of the depression subscale (defined as “moderate”) and at 1 SD (standard deviation) above the mean of the depression subscale (defined as “high”) using simple slope analyses. Gender differentially influenced outcomes among those with low and moderate levels of depression. More specifically, results indicated that at the low level of the depression subscale (HADS-D), female patients experienced asthma exacerbations more frequently, compared to male patients (aIRR = 4.30; 95% CI: 1.94–9.53). At the moderate level of the depression subscale, a similar pattern of results was observed: the expected number of asthma exacerbations among females were 82% more, compared to males (aIRR = 1.82; 95% CI: 1.07–3.13). Gender was not found to significantly influence the outcome variable at the high level of the depression subscale (aIRR = 0.40; 95% CI: 0.35–1.71). Thus, gender only influences asthma exacerbations for those who report low and moderate levels of depression. Results are presented in Table 2 and Figure 1.

## 4. Discussion

Throughout the literature, there is robust evidence of a connection between asthma and emotional disorders, including anxiety and depression, even though these findings are sometimes inconsistent. Anxiety and depression represent a common comorbidity between asthmatic patients, and clinical data has shown that the presence of psychiatric and psychological symptoms is associated with increased asthma symptoms, health service use and costs, and worsened asthma control [3,7,8]. More specifically, about 19% of asthmatic patients who are clinically stable experience clinically significant depressive symptoms, that may warrant medical intervention, and that interfere in their daily activities and self-management of their respiratory condition [14].

By performing a multivariate analysis, aimed at predicting independent factors associated with poor asthma control, Di Marco et al., found odds ratio values of 3.76 and 2.45 for anxiety and depression, respectively [15]. These values are close to those for active smoking, which is universally considered a major determinant for poor asthma control, due to the worsening of underlying bronchial inflammation [16]. In accordance with these results, our study reassures that anxiety and depression are correlated to a higher number of exacerbations per year (poorer asthma control level).

It is known that in the general population, female asthma patients are at a higher risk of anxiety and depression [17]. It is also known that patients with poorly controlled asthma are more frequently women, older, with worse pulmonary function, obese, and more anxious and/or more depressed [17]. However, the novelty in our study is that gender differentially influenced asthma outcomes among those with low, moderate, and high levels of depression. More specifically, the results indicated that at low and moderate levels of the depression subscale, female patients experienced asthma exacerbations more frequently compared to male patients, whereas gender was not found to significantly influence the outcome variable at the high level of the depression subscale.

Regarding the general population without asthma, a meta-analysis on depression diagnosis and symptoms concludes that gender differences in depression peaked in adolescence, in favor of males, but then declined and stayed stable during adulthood [18].

As far as the Greek population is concerned, there are some studies showing elderly women presenting higher rates of depression than men [19,20]. Moreover, it seems that the female gender is an independent risk factor for severe asthma exacerbation [21] and despite improved lung function and less hypercapnia, emergency hospitalization is more common in women who need longer hospital stays than men [22,23]. Combining the above data, we could conclude that women who are more susceptible to both asthma and depression would experience exacerbations more often, which is in agreement with our results.

In order to explain why there are gender differences in asthma exacerbations only in patients with low or moderate levels of depression, we should examine the connection between asthma and depression. Firstly, depression is characterized by a dysregulation of the pro-anti-inflammatory and Th1/Th2 cytokine balance [17] and has been positively associated with high systemic levels of inflammatory mediators (especially IL-4 (Interleukin-4), IL-6 and TNF-a (Tumor Necrosis Factor-a)) [24], which have been also implicated in asthma pathogenesis. Secondly, depression has known neuroendocrine effects (i.e., deregulation of the hypothalamic-pituitary-adrenocortical axis and autonomic nervous system) and has been associated with increased oxidative stress levels and decreased antioxidant functions, which may also exert a link between depression and asthma [25,26,27]. Furthermore, from a recent study from Zhang L. et al., IL-1β and TNF-α seem to serve as mediators, both for depressive symptoms, neutrophilic airway inflammation and for the impaired bronchodilator response in asthma [28].

On the other hand, findings of worse asthma control and quality of life in asthma patients with psychiatric disorders could also have occurred through cognitive or perceptual pathways, whereby psychiatric patients may have been more likely to over-report the frequency and/or severity of asthma symptoms as a result of their negative mood states [29,30]. According to the survey of Del Giacco et al., suffering from a lifetime anxiety disorder tends to significantly increase the risk of suffering a more severe form of asthma [31]. In contrast, Trojan TD et al. show that the increased likelihood of depression among patients with asthma does not appear to be exclusively related to severe or poorly controlled asthma [32].

The real question is what to do once a diagnosis of anxiety or depression is made. There is evidence that psychological interventions and pharmacological treatment are helpful in asthma patients with anxiety or depression [33,34,35]. On the other hand, we should be cautious with the use of step-up pharmacological treatment in asthma patients, who have anxiety and depression, as that may stem from their tendency to misinterpret their symptoms in favor of poorly controlled asthma. The evaluation of pulmonary function tests, inflammatory markers, such as FeNO (fraction exhaled nitric oxide) and induced sputum, can add useful information. However, such information is questionable, considering the occurrence of poorly controlled asthma with normal pulmonary function and low levels of FeNO. In such a case, close follow-up may be more advisable than the step-up of pharmacological treatment.

Finally, this is the first study to our knowledge that examines the possibility of an association between different levels of depression and asthma exacerbations. The present study may have a limitation, due to the fact that anxiety and depression have been evaluated with a simple questionnaire and not with a structured interview or with other questionnaires. However, HADS has been successfully used in several clinical groups, including asthmatic patients [3,6]. A further limitation could be the substantially small sample size of each subgroup (resulting in wide confidence intervals) and that is why results should be interpreted with caution. Further research incorporating a larger sample should be implemented to verify the moderating role of gender in the association between depression and asthma exacerbations.

## 5. Conclusions

In conclusion, the findings of this study suggest that clinicians dealing with asthmatic patients should take notice of psychological comorbidities, especially depression symptoms, during the evaluation of a patient’s health status, since these comorbidities influence exacerbation rates. Especially, when it comes to female patients with asthma, the clinician must be even more alert in order to diagnose and manage depression symptoms.

## Figures and Tables

**Figure 1 jpm-11-00586-f001:**
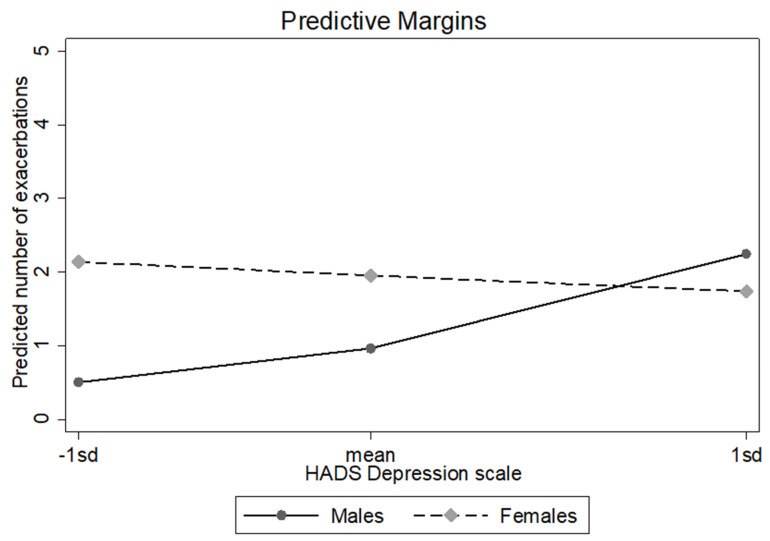
Effect of gender on exacerbations moderated by depression levels. Female patients experienced asthma exacerbations more frequently compared to male patients at the low level of the depression subscale (HADS-D).

**Table 1 jpm-11-00586-t001:** The sample’s socio-demographic and clinical characteristics.

Gender (N; %)	
Male	42 (42.00)
Female	58 (58.00)
Age (M; sd)	54.13 (12.99)
Marital status (N; %)	
Unmarried	14 (14.00)
Married	81 (81.00)
Divorced	5 (5.00)
Educational level (N; %)	
Primary schooling	16 (16.00)
Secondary schooling	47 (47.00)
Tertiary schooling	37 (37.00)
Smoking (N; %)	
Smoker	26 (26.00)
Ex-smoker	30 (30.00)
Non-smoker	44 (44.00)
Time since diagnosis (in years) (Mdn; IQR)	13.00 (13.00)
HADS total score (Mdn; IQR)	8.00 (7.00)
HADS-Anxiety score (Mdn; IQR)	4.00 (4.00)
HADS-Depression score (Mdn; IQR)	4.00 (2.00)

N: Number, M: mean, sd: standard deviation, Mdn: median, IQR: interquartile range, HADS: Hospital Anxiety Depression Scale.

**Table 2 jpm-11-00586-t002:** Results from negative binomial regression model.

Variable	IRR	95% CI	*p*-Value
HADS Anxiety score ^ⱡ^	1.04	0.97–1.12	0.238
HADS Depression score ^ⱡ^	1.33	1.10–1.62	0.004
Females	1.82	1.06–3.12	0.028
HADS Depression score × Females	0.72	0.58–0.90	0.004

^ⱡ^ Both HADS Anxiety and HADS Depression scores were centered at their means. Note: IRR: Incidence Rate Ratio; CI: 95% Confidence Intervals.

## Data Availability

Data sheet is available in the form of Excel after contact with the corresponding author.

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
