# Peer review of "Depression Levels Influence the Rate of Asthma Exacerbations in Females"

_jpm, 2021, doi:10.3390/jpm11060586_

Round 1

Reviewer 1 Report

Anastasia et al. studied interaction effects between gender and depression subscale among one hundred patients with asthma in Greece. The authors have made significant progress towards individualized asthma treatment strategies by evaluating the clinical implications of depression and anxiety in asthma patients from various angles.

It would be better to address some concerns for scientific soundness before the acceptance of the manuscript.

  1. In women, stress responses depending on menopause may appear differently, so adding information about it will be helpful if possible.
  2. Please describe specifically because the severity of the HADS scale is not provided in the methods.
  3. Please provide information on the number of patients initially enrolled and excluded. Furthermore, how was the assessment of diseases that met the exclusion criteria?
  4. In the manuscript, it appears that no tables or pictures are inserted. Please check and present them in the appropriate location.

Author Response

We appreciate the reviewer’s attention to detail in our manuscript.

  1. In our study the percentage of women, who were above 50 years old is 67%. However, no information about menopausal status is available.
  2. A further description of HADS scale has been added in the section “Materials and Methods”: “A total subscale score of >8 points out of possible 21 points, indicates significant symptoms of anxiety or depression”.
  3. We have enrolled 104 patients. From them 4 were excluded due to lost in follow up. The assessment of diseases that met the exclusion criteria has been performed by two experienced clinicians. In detail we have been added in the section “Materials and Methods” the following sentence: “Two experienced clinicians have been taken detailed personal and family history of all participants”.
  4. Thank you for your comment. In our manuscript, two tables and 1 figure are included with special reference in the section “Results”. For your information, the tables and figure are presented bellow:

Table 1. Sample’s socio-demographic and clinical characteristics.

Gender (N; %)

Male

42 (42.00)

Female

58 (58.00)

Age (M; sd)

54.13 (12.99)

Marital status (N; %)

Unmarried

14 (14.00)

Married

81 (81.00)

Divorced

5 (5.00)

Educational level (N; %)

Primary schooling

16 (16.00)

Secondary schooling

47 (47.00)

Tertiary schooling

37 (37.00)

Smoking (N; %)

Smoker

26 (26.00)

Ex-smoker

30 (30.00)

Non-smoker

44 (44.00)

Time since diagnosis (in years) (Mdn; IQR)

13.00 (13.00)

HADS total score (Mdn; IQR)

8.00 (7.00)

HADS-Anxiety score (Mdn; IQR)

4.00 (4.00)

HADS-Depression score (Mdn; IQR)

4.00 (2.00)

N: Number, M: mean, sd: standard deviation, Mdn: Median, IQR: Interquartile Range, HADS: Hospital Anxiety Depression Scale.

Table 2. Results from negative binomial regression model.

Variable

IRR

95% CI

p-value

HADS Anxiety scoreⱡ

1.04

0.97-1.12

0.238

HADS Depression scoreⱡ

1.33

1.10-1.62

0.004

Females

1.82

1.06-3.12

0.028

HADS Depression score x Females

0.72

0.58-0.90

0.004

ⱡBoth HADS Anxiety and HADS Depression scores were centered at their means.

Note: IRR: Incidence Rate Ratio; CI: 95% Confidence Intervals.

Figure 1. (legend). Effect of gender on exacerbations moderated by depression levels. Female patients experienced asthma exacerbations more frequently compared to male patients at the low level of the Depression subscale (HADS-D).

Reviewer 2 Report

The manuscript was well written except for a minor word or two missing in the conclusion of the abstract and  the only suggestion that I have for the authors and add a descriptor to the y-axis (vertical).

Author Response

The reviewer’s comment is correct: The structure of the abstract (bold-italics) has been corrected.

The descriptor for the y-axis is: “Predicted number of exacerbations”. The descriptor has been added to the figure 1.

Figure 1. (legend). Effect of gender on exacerbations moderated by depression levels. Female patients experienced asthma exacerbations more frequently compared to male patients at the low level of the Depression subscale (HADS-D).

Reviewer 3 Report

The article presents the well-known problem of asthma associated with depression. He could also assess anxiety, which is often associated with similar questionnaires. The article does not present relevant or new data for the literature, it did not perform an analysis according to the severity of asthma or asthma control. Although it is relatively well written, the group of patients is small, and the results do not bring new and relevant information for the reader

Author Response

With respect to reviewer’s comments, indeed the sample of patients is relatively small. An analysis according to the severity of asthma and asthma control has been performed. Respectively, ACT score was negatively associated with HADS score (R = -0.339, p = 0.0395). However, our attempt was to bring something novel in the medical field. Thus, we focused on our main finding that was that the gender differentially influenced asthma outcomes among those with low, moderate and high levels of depression.

Round 2

Reviewer 2 Report

The additional input was helpful to improve/clarify the manuscript.  I have no additional suggestions to improve this manuscript.

Reviewer 3 Report

Could be publishe